# A Modified Constitutive Model for the Description of the Flow Behavior of the Ti-10V-2Fe-3Al Alloy during Hot Plastic Deformation

**Fukang Wang, Jingyuan Shen, Yong Zhang and Yongquan Ning \***

School of Materials Science and Engineering, Northwestern Polytechnical University, Xi'an 710072, China
\* Correspondence: luckyning@nwpu.edu.cn; Tel.: +86-29-8849-2642

**Abstract:** The hot deformation behavior of the aerospace Ti-10-2-3 alloy was investigated by isothermal compression tests at temperatures of 740 to 820 °C and strain rates of 0.0005 to 10 s$^{-1}$. The results show that the studied alloy is extremely sensitive to deformation parameters, like the temperature and strain rate. The temperature mainly affects the magnitude of flow stress at larger strains, while the strain rate not only affects the value of flow stress but also the shape of the flow curves. At low strain rates, the flow stress increases with strain, followed by a broad peak and then remains almost constant. At high strain rates, the flow curves exhibit a hardening to a sharp peak at small strains, followed by a rapid dropping to a plateau caused by dynamic softening. In order to describe such flow behavior, a constitutive model considering the effect of deformation parameters was developed as an extension of an existing constitutive model. The modified constitutive model (MC) was obtained based on the original constitutive model (OC) by introducing a new parameter to compensate for the error between the experimental data and predicted values. Compared to the original model, the developed model provides a better description of the flow behavior of Ti-10-2-3 alloy at elevated temperatures over the specified deformation domain.

**Keywords:** titanium alloys; hot deformation; flow behavior; constitutive model

## 1. Introduction

High strength and high toughness titanium alloys have been widely utilized for the fabrication of landing gears' structure in the aerospace industry due to its high specific strength, excellent fracture toughness, and fatigue resistance [1,2]. As one of the most widely used near-β titanium alloys, Ti-10-2-3 was developed as a forging alloy for applications in the aerospace industry owing to its good combination of high strength and good toughness [3].

The finite element analysis technique has been extensively used as an effective numerical tool in forming process optimization and structural design. The effectiveness of numerical simulations is remarkably affected by the reliability and accuracy of the predictability of the inputted constitutive model [4,5]. In this context, it is of great significance to establish constitutive models with strong applicability and high accuracy. However, the mechanical response of titanium alloy is greatly sensitive to the specified experimental conditions, like the temperature and strain rate, and the strain [6]. Using common constitutive models, it is difficult to describe such a behavior. Therefore, the construction of accurate constitutive model becomes very important [7].

A considerable effort of constitutive modelling has been carried out to describe the hot deformation behavior of metals and alloys over the last decades. The developed constitutive models are generally divided into three categories: Phenomenological, physically-based, and artificial neural network (ANN) models [8]. A phenomenological model proposed by Sellars and Tegart [9], in which the flow stress is

represented by the sine-hyperbolic law of the Arrhenius model, has been widely used to describe the hot deformation behavior of metallic materials. Lin et al. [10] proposed a modified Arrhenius model to characterize the correlations between the flow stress, strain rate, and temperature of 42CrMo steel at high temperatures through the compensation of strain and the strain rate. Some other phenomenological models, like Johnson-Cook (JC), modified Johnson-Cook (MJC), modified Zerilli-Armstrong (MZA), Fields Backofen (FB), and modified Fields Backofen (MFB), have successfully been used to describe the hot flow behaviors of 20CrMo alloy [11], Ti-6Al-4V alloy [12], Ti-6Cr-5Mo-5V-4Al alloy [13], AZ31 magnesium alloy [14], and AZ31B magnesium alloy [15], respectively. The above-mentioned models are widely used for predicting flow stress at the post-peak stage but are seldom applied to express the flow behavior at the prior peak region, especially for flow curves with a sharp peak at a small strain. In this case, physically-based models and ANN models are considered as effective methods to model the flow curve with a sharp peak. Zhang et al. [16] set up a physically-based constitutive model to describe the flow behavior of a high strength aluminum alloy in hot working conditions. The Bammann-Chiesa-Johnson (BCJ) [17] and mechanical threshold stress (MTS) [18] models are also applied to deformed metallic materials with a satisfactory accuracy. The limitation of physically-based models is that they need more parameters, which have to be calculated from the tested data through a complex algorithm. ANN models based on biological neural networks have been widely applied for the prediction of hot deformation behaviors in recent years due to their simplification of the process of postulating the model and determining parameters [19]. Further, they have been applied successfully for the description or simulation of flow behaviors of aluminum alloy [20], magnesium alloy [21], and titanium alloy [22]. However, the disadvantage of the ANN model is that it cannot provide a specific computational description that can be utilized later. Therefore, a perfect constitutive model must contain an appropriate number of material parameters and predict the flow behavior under all tested stages accurately and reliably by using limited experimental data.

Within this study, the hot deformation behavior of Ti-10-2-3 alloy was studied by conducting an isothermal compression experiment in the temperature range of 740 to 820 °C and strain rates of 0.0005 to 10 s$^{-1}$. Based on the experimental data and the characteristics of the flow curves, the true stress-strain curves of the Ti-10-2-3 alloy were segmented into the prior-peak and post-peak stages by taking the peak as the demarcation point. Based on this segmentation, a modified constitutive model (MC) was developed as an extension of the original constitutive model (OC) by introducing a new parameter. The suitability of applying the OC and MC model for the description of the flow behavior of titanium alloy was evaluated by comparing the experimental and predicted data.

## 2. Experiments

The chemical compositions (wt.%) of the investigated titanium alloy were as follows: Al 2.8, Fe 1.85, V 10.41, and Ti bal. Cylindrical specimens with a diameter of 8 mm and a height of 12 mm were machined from the wrought billet. The specimens were compressed isothermally on a Gleebe-1500D thermo-mechanical simulator at five sets of deformation temperatures (740, 760, 780, 800, and 820 °C) and six sets of strain rates (0.0005, 0.001, 0.01, 0.1, 1, and 10 s$^{-1}$) to a height reduction of 60%. Tantalum foil with a thickness of 0.1 mm was used in order to reduce the friction between the workbench and the specimen to minimize the occurrence [6]. The specimens were firstly preheated to 920 °C at a heating rate of 10 °C/s and held for 3 min, then cooled at 10 °C/s to the deformation temperature and soaked for 0.5 min to eliminate thermal gradients. The stress-strain curves were recorded automatically.

Typical true stress-strain curves of the studied titanium alloy under the deformation temperatures of 740 and 760 °C and the strain rates of 0.0005 and 1 s$^{-1}$ are shown in Figure 1. It is clear that the flow curve under the low strain rate of 0.0005 s$^{-1}$ presents a broad peak and then smoothly transitions to the steady state (curve a). A nearly identical curve, which is not shown in Figure 1, was obtained at 760 °C and 0.0005 s$^{-1}$. In contrast, at a high strain rate of 1 s$^{-1}$, the flow curve shows a sharp peak (point A). After that peak, the flow curve rapidly drops to a plateau (i.e., discontinuous yield characteristic and the yield point of B) caused by dynamic recrystallization (DRX), which is shown

in curve c. Comparing curves b and c, the shapes of the flow curves are similar despite the different deformation temperatures. The essential difference in curve a and curves b and c indicates that the temperature affects the magnitude of flow stress, while the strain rate not only affects the value of the flow stress but also determines the shape of the flow curves significantly.

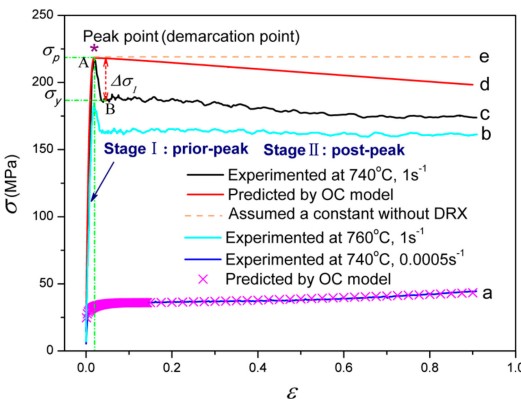

**Figure 1.** Typical true stress-true strain curve.

## 3. Theories

In order to describe the flow curve behavior shown in Figure 1, McQueen et al. [23] proposed the equation:

$$\frac{\sigma}{\sigma_p} = \left[\frac{\varepsilon}{\varepsilon_p} \exp\left(1 - \frac{\varepsilon}{\varepsilon_p}\right)\right]^c, \tag{1}$$

where $\sigma_p$ and $\varepsilon_p$ are the peak stress and peak strain, respectively. Further, $c$ is a material constant, which depends on the strain rate, $\dot{\varepsilon}$, and deformation temperature, $T$. A necessary step to characterize $\sigma_p$ and $\varepsilon_p$ is to access the calculation procedure of the Arrhenius equation. From the characteristics of Equation (1), the flow curve can be divided into the prior-peak stage (stage I) and the post-peak stage (stage II) by taking the peak as the demarcation point. Noticeably, Equation (1) was initially used for the description of the flow behavior of the austenitic stainless steels 301, 304, and 317 at the prior-peak stage [24–26], where the flow stress can be assumed as constant when the strain exceeds peak strain (stage II), as shown in curve e. Lin et al. [27] extended the application of this model from work-hardening (WH) to all the deformation stages by evaluating the values of $c$ in the WH and dynamic softening (DS) stages separately. The obtained shape of the predicted flow curves of the Ni-based superalloy was similar to that of the austenitic steels, which reveals a wide peak range. Hence, it is necessary to explore the suitability of the studied model for titanium alloys.

In the current work, Equation (1) was firstly applied to the Ti-10-2-3 alloy and its application scope was extended from stage I to all the deforming stages.

## 4. Results

Based on Equation (1), the flow behavior of Ti-10-2-3 will be modelled in the prior- and the post-peak stage according to the approach of Lin et al. [27] and Hajari et al. [28].

### 4.1. Modelling the Flow Behavior in the Prior-Peak Stage

Generally, the discussed constitutive model is applied to predict the flow stress in the prior-peak stage. Based on Equation (1), the flow stress under stage I can be expressed as [27]:

$$\frac{\sigma}{\sigma_p} = \left[\frac{\varepsilon}{\varepsilon_p} \exp\left(1 - \frac{\varepsilon}{\varepsilon_p}\right)\right]^{c_1}, \tag{2}$$

where $c_1$ is a material constant.

From Equation (2), obviously, the peak stress and peak strain should be primarily obtained to achieve the predicted stress. In general, the peak stress can be evaluated by the hyperbolic-sine Arrhenius-type constitutive equation, which is given by [28]:

$$\dot{\varepsilon} = AF(\sigma_{\mathrm{p}}) \exp\left(-\frac{Q}{RT}\right), \text{ where } F(\sigma_{\mathrm{p}}) = \begin{cases} \sigma_{\mathrm{p}}^{n'} & \alpha\sigma_{\mathrm{p}} < 0.8 \\ \exp(\beta\sigma_{\mathrm{p}}) & \alpha\sigma_{\mathrm{p}} > 1.2 \\ [\sinh(\alpha\sigma_{\mathrm{p}})]^{n} & \text{for all } \sigma_{\mathrm{p}} \end{cases}, \tag{3}$$

where $\dot{\varepsilon}$ is the strain rate; $Q$ is the activation energy of hot deformation (J·mol$^{-1}$); $R$ is the universal gas constant (8.314 J·mol$^{-1}$·K$^{-1}$); $T$ is the absolute temperature (K); and $A$, $n'$, $\alpha$, $\beta$, and $n$ are material constants, whereby $\alpha = \beta/n'$.

It is well known that the combined effects of the deformation temperature and strain rate on flow behavior can be expressed by the Zener-Hollomon parameter in an exponent-type equation, which is defined as [29]:

$$Z = \dot{\varepsilon} \exp\left(\frac{Q}{RT}\right) = A[\sinh(\alpha\sigma_{\mathrm{p}})]^{n}. \tag{4}$$

The flow stress can be calculated as a function of the Zener-Hollomon parameter as follows [30]:

$$\sigma_{\mathrm{p}} = \frac{1}{\alpha} \ln\left\{ \left(\frac{Z}{A}\right)^{1/n} + \left[\left(\frac{Z}{A}\right)^{2/n} + 1\right]^{1/2} \right\}. \tag{5}$$

Substituting $F(\sigma_{\mathrm{p}})$ in Equation (3) and taking the nature logarithm on both sides yields:

$$\ln(\sigma_{\mathrm{p}}) = \frac{1}{n'}\ln(\dot{\varepsilon}) - \frac{1}{n'}\ln(B), \text{ with } B = A_1 \exp\left(-\frac{Q}{RT}\right), \tag{6}$$

$$\sigma_{\mathrm{p}} = \frac{1}{\beta}\ln(\dot{\varepsilon}) - \frac{1}{\beta}\ln(C), \text{ with } C = A_2 \exp\left(-\frac{Q}{RT}\right), \tag{7}$$

$$\ln[\sinh(\alpha\sigma_{\mathrm{p}})] = -\frac{1}{n}\ln A + \frac{1}{n}\ln\dot{\varepsilon} + \frac{Q}{nRT}, \tag{8}$$

where $A_1$, $A_2$, $B$, and $C$ are the material constants.

Substituting the flow stress and strains into Equations (6) and (7), the constants, $n'$ and $\beta$, can be obtained from the line's slopes of the $\ln(\sigma_{\mathrm{p}}) - \ln(\dot{\varepsilon})$ and $\sigma_{\mathrm{p}} - \ln(\dot{\varepsilon})$ plots, respectively. The mean values of $n'$ and $\beta$ were found to be 4.4962 and 0.04692 MPa$^{-1}$. Hence, $\alpha = \beta/n' = 0.01044$ MPa$^{-1}$. Substituting the obtained values of $\alpha$, peak stress, and strain rate into Equation (8), the relation between $\ln[\sinh(\alpha\sigma_{\mathrm{p}})]$ and $\ln(\dot{\varepsilon})$ is shown in Figure 2c. Therefore, the constant, $n$, can be obtained in a similar way and was found to be 3.2599.

In Equation (8), it is assumed that the effects of the strain rate and deformation temperature on activation energy are two independent variables. For the given strain rate and strain condition, taking the partial differential with respect to $T^{-1}$ in Equation (8) yields:

$$Q = nR \left.\frac{\partial \ln[\sinh(\alpha\sigma_{\mathrm{p}})]}{\partial\left(\frac{1}{T}\right)}\right|_{\dot{\varepsilon}}. \tag{9}$$

Obviously, the value of $Q$ can be obtained from the average slope of the lines in the $\ln[\sinh(\alpha\sigma_{\mathrm{p}})]$-$T^{-1}$ plot as shown in Figure 2d and this was evaluated as 240.005 KJ·mol$^{-1}$. Substituting $\alpha$, $n$, $Q$, and other deformation parameters, like $T$ and $\dot{\varepsilon}$, into Equation (4), the value of $A$ can be determined from the intercept of the regression line of the $\ln Z$-$\ln[\sinh(\alpha\sigma_{\mathrm{p}})]$ plot (Figure 2e) and was found to be $2.2035 \times 10^{10}$ s$^{-1}$.

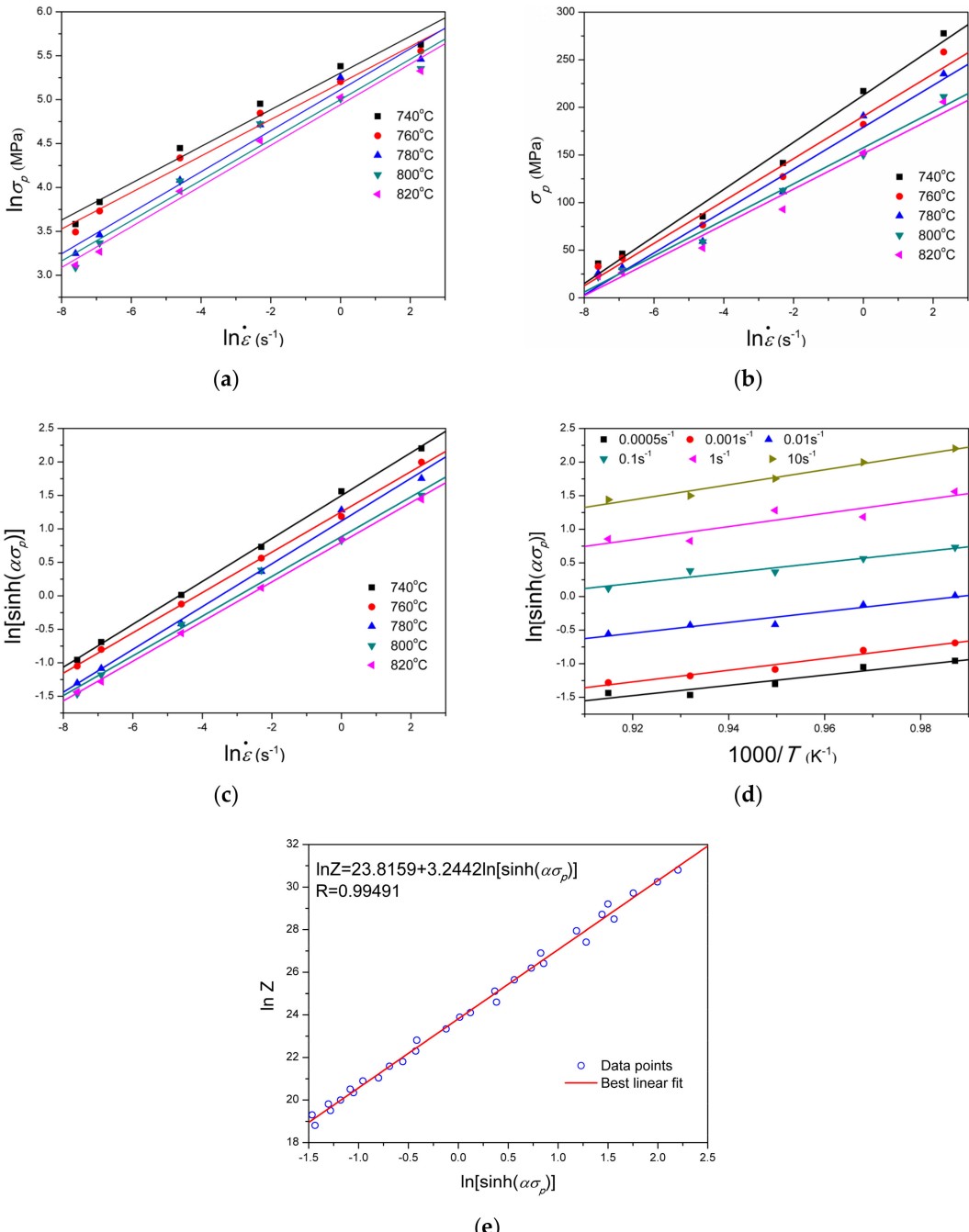

**Figure 2.** Evaluating the value of (**a**) $n'$ by plotting $\ln\sigma_p$ vs. $\ln\dot{\varepsilon}$; (**b**) β by plotting $\sigma_p$ vs. $\ln\dot{\varepsilon}$; (**c**) $n$ by plotting $\ln[\sinh(\alpha\sigma_p)]$ vs. $\ln\dot{\varepsilon}$; (**d**) $Q$ by plotting $\ln[\sinh(\alpha\sigma_p)]$ vs. $1000/T$; and (**e**) $A$ by plotting $\ln Z$ vs. $\ln[\sinh(\alpha\sigma_p)]$.

For modelling, it is useful to express the peak strain as a function of the Zener-Hollomon parameter, which can be expressed as the following equation:

$$\varepsilon_{\mathrm{P}} = kZ^q, \tag{10}$$

where $k$ and $q$ are material constants. Taking the natural logarithm on both sides of Equation (10):

$$\ln\varepsilon_{\mathrm{P}} = q\ln Z + \ln k, \tag{11}$$

the values of $k$ and $q$ can be evaluated from the y-intercept and the slope of the regression line in the $\ln\varepsilon_p$-$\ln Z$ plot (Figure 3), which were evaluated as $5.818 \times 10^{-4}$ and 0.12, respectively.

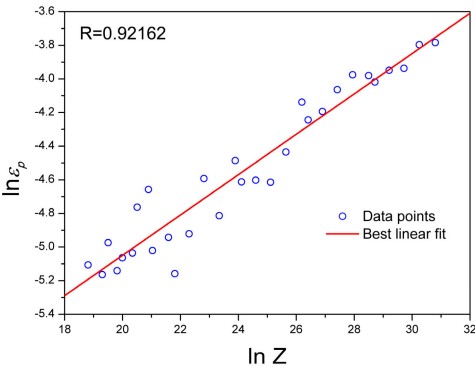

**Figure 3.** Relation between peak strain and Zener–Hollomon parameter.

Based on Equations (4), (5), and (10), the peak stress and peak strain of the studied Ti-10-2-3 alloy can now be described as a function of the parameter, $Z$:

$$\begin{cases} \sigma_p = \frac{1}{0.01044} \ln\left\{ \left( \frac{Z}{2.2035 \times 10^{10}} \right)^{\frac{1}{3.26}} + \left[ \left( \frac{1}{2.2035 \times 10^{10}} \right)^{\frac{2}{3.26}} + 1 \right]^{\frac{1}{2}} \right\} \\ \varepsilon_p = (5.818 \times 10^{-4}) Z^{0.12} \\ Z = \dot{\varepsilon} \exp\left( \frac{240005}{RT} \right) \end{cases} \tag{12}$$

In the following section, the effects of the deformation temperature and strain rate on the material constant, $c_1$, are investigated. The procedure of determining the material constant, $c_1$, is shown for the deformation temperature of 740 °C and strain rate of 0.0005 s$^{-1}$ as an example. For the determination of $c_1$, taking the natural logarithm on both sides of the Equation (2), one gains:

$$\ln\left( \frac{\sigma}{\sigma_p} \right) = c_1 \left[ \ln\left( \frac{\varepsilon}{\varepsilon_p} \right) + \left( 1 - \frac{\varepsilon}{\varepsilon_p} \right) \right]. \tag{13}$$

Substituting the peak stress and peak strain from Equation (12) as well as the values of the measured stress and strain during the prior-peak stage into Equation (13), the value of $c_1$ can be determined from the slope of the regression line of the plot of $\ln(\sigma/\sigma_p)$-$[\ln(\varepsilon/\varepsilon_p) + (1 - \varepsilon/\varepsilon_p)]$, as shown in Figure 4. Remarkably, the regression line is in good agreement with the experimental scattered points at the temperature of 740 °C and the strain rate of 0.0005 s$^{-1}$. Accordingly, the material constant, $c_1$, can easily found to be 0.1014 from the regression line.

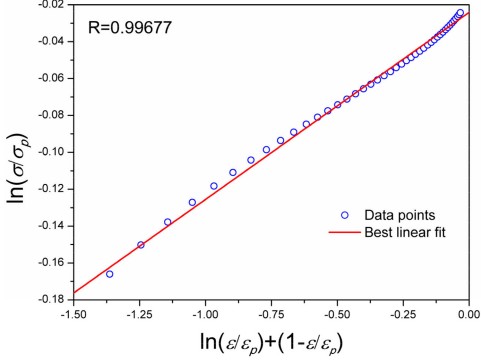

**Figure 4.** Relation between $\ln(\sigma/\sigma_p)$ and $[\ln(\varepsilon/\varepsilon_p) + (1 - \varepsilon/\varepsilon_p)]$ in the prior-peak stage.

Repeating this calculation procedure for each tested combination of the temperature and strain rate, the values of the material constant, $c_1$, given in Table 1 were found. Obviously, the value of $c_1$ varies with the deformation temperature and strain rate, which indicates that it strongly depends on deformation parameters. It may be that a reasonable way to construct the relation between the material constant, $c_1$, and the Zener-Hollomon parameter is to plot $\ln c_1$-$\ln Z$, as shown in Figure 5. Through a liner regression analysis, the y-intercept and the slope of the regression line in the plot of $\ln c_1$-$\ln Z$ can be evaluated as −2.6451 and 0.0903, respectively. The material constant, $c_1$, under different deformation conditions can then be expressed as:

$$c_1 = \exp(-2.6451)Z^{0.0903} = 0.071 Z^{0.0903}. \tag{14}$$

**Table 1.** The values of the material constant, $c_1$, for all tested loading combinations.

| Strain Rate (s$^{-1}$) | Deformation Temperature (°C) | | | | |
|---|---|---|---|---|---|
| | **740** | **760** | **780** | **800** | **820** |
| 0.0005 | 0.10143 | 0.10143 | 0.55065 | 0.34829 | 0.31172 |
| 0.001 | 0.42785 | 0.35682 | 0.25596 | 0.35636 | 0.29254 |
| 0.01 | 0.72538 | 0.71803 | 0.62142 | 0.58776 | 0.75455 |
| 0.1 | 0.89752 | 0.80238 | 1.08625 | 0.72074 | 1.04578 |
| 1 | 0.86542 | 1.09427 | 1.09299 | 0.98385 | 0.96104 |
| 10 | 0.86736 | 0.85978 | 0.56476 | 0.81574 | 0.80480 |

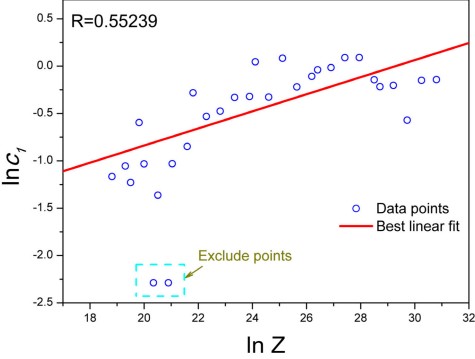

**Figure 5.** Relation between the material constant, $c_1$, and the Zener-Hollomon parameter.

### 4.2. Modelling the Flow Behavior in the Post-Peak Stage

As mentioned, the original constitutive model was used for the description of the flow behavior in the prior-peak stage. Lin et al. [27] revised the model and successfully extended its application to all deformation stages (including work hardening and dynamic softening stages) of an Ni-based superalloy. In this section, the purpose is to explore the suitability of the equation:

$$\frac{\sigma}{\sigma_p} = \left[ \frac{\varepsilon}{\varepsilon_p} \exp\left( 1 - \frac{\varepsilon}{\varepsilon_p} \right) \right]^{c_2}, \tag{15}$$

at large strains, where $c_2$ is a material constant.

With the same approach as that already shown in Section 4.1, a nearly linear relation between $\ln(\sigma/\sigma_p)$ and $[\ln(\varepsilon/\varepsilon_p) + (1 - \varepsilon/\varepsilon_p)]$ at a temperature of 740 °C and strain rate of 0.001 s$^{-1}$ was obtained, as shown in Figure 6. However, for higher strain rates, such as 0.1, 1, and 10 s$^{-1}$, the data points are randomly distributed in the plot of $\ln(\sigma/\sigma_p)$-$[\ln(\varepsilon/\varepsilon_p) + (1 - \varepsilon/\varepsilon_p)]$. Assuredly, the deviation degree between the scattered points and the regression line remarkably depends on the range of the strain. Thus, a suitable strain interval may derive the constant, $c_2$, with a high confidence level. Correspondingly, the predicted flow stress in this interval will match the experimental ones well. The

effect of strain on $c_2$ was verified in the investigation accomplished by Lin et al. [27]. However, the present study mainly focused on elevating the predictability of the model in the large strain range rather than discussing the suitability of the partial strain region.

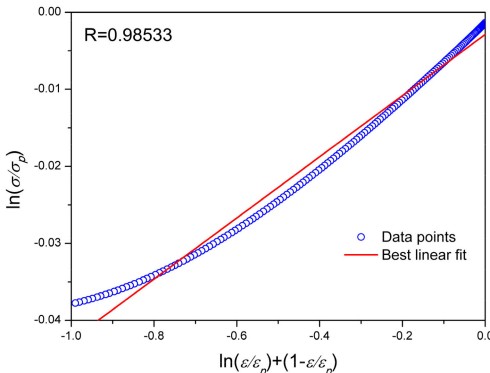

**Figure 6.** Relation between $\ln(\sigma/\sigma_p)$ and $[\ln(\varepsilon/\varepsilon_p) + (1 - \varepsilon/\varepsilon_p)]$ in the post-peak stage.

Accordingly, the peak stress and the peak strain under all of the tested deformation conditions can be obtained on the basis of Equation (12). Repeating the calculation procedure for each tested combination of the temperature and strain rate, the values of the material constant, $c_2$, given in Table 2 were found. The relation between the material constant, $c_2$, the deformation temperature, and the strain rate can be derived from the correlation between $c_2$ and the Zener-Hollomon parameter by plotting $\ln c_2$ vs. $\ln Z$, as shown in Figure 7. The $y$-intercept and the slope of the regression line can be derived as $-9.9310$ and $0.1279$, respectively. So, the material constant $c_2$ can be described as:

$$c_2 = \exp(-9.9310)Z^{0.1279} = (4.86 \times 10^{-5})Z^{0.1279}. \tag{16}$$

**Table 2.** The values of the material constant, $c_2$, for all of the tested loading combinations.

| Strain Rate (s$^{-1}$) | Deformation Temperature (°C) | | | | |
|---|---|---|---|---|---|
| | **740** | **760** | **780** | **800** | **820** |
| 0.0005 | 0.01268 | 0.00102 | 0.00412 | $6.08 \times 10^{-4}$ | $8.65 \times 10^{-4}$ |
| 0.001 | $7.27 \times 10^{-4}$ | 0.00126 | 0.00145 | 0.00109 | $3.28 \times 10^{-4}$ |
| 0.01 | $3.89 \times 10^{-4}$ | $4.15 \times 10^{-4}$ | $3.14 \times 10^{-4}$ | $5.99 \times 10^{-4}$ | 0.00133 |
| 0.1 | 0.00221 | 0.00163 | 0.00138 | $6.45 \times 10^{-4}$ | 0.00148 |
| 1 | 0.00221 | $5.39 \times 10^{-4}$ | $7.07 \times 10^{-4}$ | $7.04 \times 10^{-4}$ | 0.00234 |
| 10 | 0.00608 | 0.00609 | 0.00362 | 0.002 | 0.00136 |

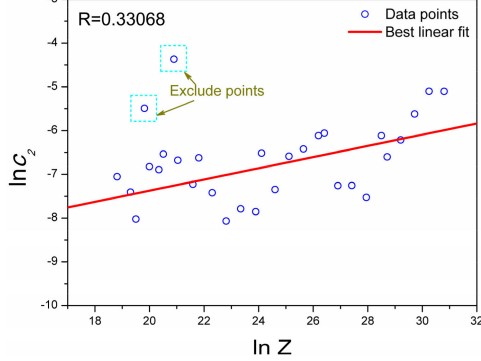

**Figure 7.** Relation between the material constant, $c_2$, and the Zener-Hollomon parameter.

### 4.3. Application of the Original Constitutive Model

Based on the parameter identification in the prior-peak stage (Section 4.1) and the post-peak stage (Section 4.2), a constitutive model was established:

$$\begin{cases} \frac{\sigma}{\sigma_\text{p}} = \left[\frac{\varepsilon}{\varepsilon_\text{p}}\exp\left(1 - \frac{\varepsilon}{\varepsilon_\text{p}}\right)\right]^c \\ \sigma_\text{p} = \frac{1}{0.01044}\ln\left\{\left(\frac{Z}{2.2035\times10^{10}}\right)^{\frac{1}{3.26}} + \left[\left(\frac{Z}{2.2035\times10^{10}}\right)^{\frac{2}{3.26}} + 1\right]^{\frac{1}{2}}\right\} \\ \varepsilon_\text{p} = (5.818\times10^{-4})Z^{0.12} \\ Z = \dot{\varepsilon}\exp\left(\frac{240005}{RT}\right) \\ c = \begin{cases} 0.071Z^{0.0903}(\varepsilon \le \varepsilon_\text{p}) \\ (4.86\times10^{-5})Z^{0.1279}(\varepsilon > \varepsilon_\text{p}) \end{cases} \end{cases} \quad (17)$$

In order to verify the constitutive model, a comparison between the experimental and the predicted flow stress of Ti-10-2-3 alloy in a wide range of deformation temperatures, strain rates, and strain are shown in Figure 8. It is clear that the predicted stress values accurately reflect the flow behavior of the studied alloy in the prior-peak stage under all the deformation conditions. Further, there is a good correlation between the predicted data and the experimental ones for all deformation temperatures at low strain rates, like 0.0005 and 0.001 s$^{-1}$. However, some larger deviations between the predicted data and experimental ones can be noticed at high strain rates (1 and 10 s$^{-1}$), especially when the flow curve exhibits a sharp peak and then drops rapidly to a plateau. The reason is that the dislocation multiplication by the initial work-hardening overcomes the dislocation annihilation by dynamic recovery, which provides a driving force that evokes a 'surge' in the dynamic recovery rate of the β phase, resulting in the flow stress dropping dramatically [31]. However, there is a smooth transition from the peak to the steady state during the formation at low strain rates (0.0005 and 0.001 s$^{-1}$). The work-hardening effect initially predominates and is subsequently insufficient to offset the dynamic recovery effect since the low strain rate guarantees sufficient time for accumulating thermal activation energy and migrating the nucleation and growth of dynamically recrystallized grains at the boundaries [32]. Further, elevated temperatures promote the occurrence of thermally-activated cross-slip and an increase of dislocations. This softening effect cancels out the preliminary hardening effect, and the value of flow stress is essentially unchanged with the increasing strain.

An interesting phenomenon observed from Figure 8 is that the peak's shape has little effect on the predictability of the constitutive model in the prior-peak stage, while the predictability in the post-peak stage significantly relies on the peak's shape. Namely, the deviation increases as the fall from the peak to the steady state increases. Meanwhile, as we can see in Figure 8, the pre-calculated curve is nearly parallel to the tested curve; the value of deviation is approximately equal to $\Delta\sigma_1 = \sigma_\text{p} - \sigma_\text{y}$ as shown in Figure 1 ($\sigma_\text{p}$ and $\sigma_\text{y}$ are the peak stress and yield stress, respectively). In this case, the prediction error induced by the difference between the peak and plateau is inevitable. In addition, the value of the intercept in the $\ln(\sigma/\sigma_\text{p})$-$[\ln(\varepsilon/\varepsilon_\text{p}) + (1 - \varepsilon/\varepsilon_\text{p})]$ plot is smaller than zero (take the real intercept value as 0), which does not consider the real value of the intercept in Figure 6. This causes the value of $c$ to be greater than the true value. So, the predicted values are greater than the experimental values, as shown in Figure 8. Hence, the idea is to introduce a new parameter into Equation (2), which helps to minimize error between the predicated and the experimental data.

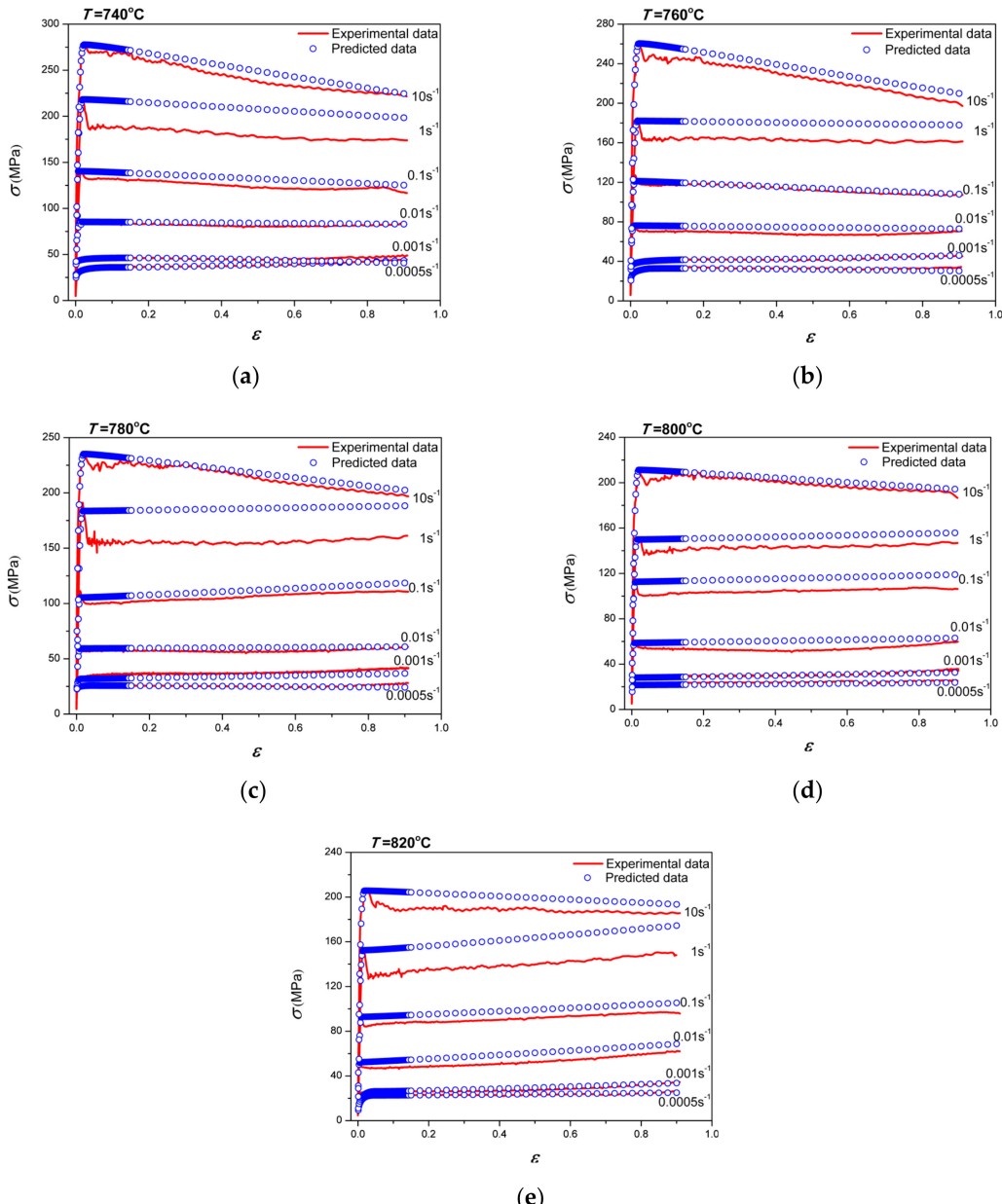

**Figure 8.** Comparisons between the experimental and predicted flow stress at deformation temperatures of: (**a**) 740 °C; (**b**) 760 °C; (**c**) 780 °C; (**d**) 800 °C; and (**e**) 820 °C.

### 4.4. Development of a Modified Constitutive Model

As seen in Section 4.3, the original constitutive model (OC) could not describe the measured flow curve with adequate accuracy when the flow curve exhibited a sharp peak. In order to be able to describe the flow curve over the whole strain range, a new parameter, $\zeta$, is introduced into Equation (2) to ameliorate the limitation of the predictability for the studied model. The modified constitutive model (MC) can be written as:

$$\frac{\sigma}{\sigma_{\mathrm{p}}} = \zeta \cdot \left[ \frac{\varepsilon}{\varepsilon_{\mathrm{p}}} \exp\left(1 - \frac{\varepsilon}{\varepsilon_{\mathrm{p}}}\right) \right]^{c}, \tag{18}$$

in which $\zeta$ is a material constant.

Taking the natural logarithm on both sides of Equation (18) yields:

$$\ln\left(\frac{\sigma}{\sigma_P}\right) = c \cdot \left[\ln\left(\frac{\varepsilon}{\varepsilon_P}\right) + \left(1 - \frac{\varepsilon}{\varepsilon_P}\right)\right] + \ln \zeta. \tag{19}$$

With the same procedure as in Sections 4.1 and 4.2, the relation between $\ln(\sigma/\sigma_P)$ and $[\ln(\varepsilon/\varepsilon_P) + (1 - \varepsilon/\varepsilon_P)]$ is established, which is referenced as Figures 4 and 6. The intercepts (i.e., $\ln\zeta$) of these two plots can be found as 0.97618 and 0.99713, respectively. By repeating this procedure for all loading combinations, the values given in Tables 3 and 4 were obtained for $\zeta_1$ in the prior-peak and $\zeta_2$ in the post-peak stage, respectively. The values of $\zeta$ in the two stages strongly depends on the shape of the flow curve, especially in the post-peak stage. The relations of the material constant, $\zeta$, the deformation temperature, $T$, and the strain rate, $\dot{\varepsilon}$, can be established from the $\ln\zeta$-$\ln Z$ plot, as shown in Figure 9. Hence, the material constants, $\zeta_1$ and $\zeta_2$, can be expressed as a function of the parameter, $Z$:

$$\zeta_1 = 1.035 Z^{-0.00263}, \text{ and } \zeta_2 = 1.141 Z^{-0.00741}. \tag{20}$$

**Table 3.** The values of the material constant, $\zeta_1$, under all of the tested specified loading conditions.

| Strain Rate (s$^{-1}$) | Deformation Temperature (°C) | | | | |
|---|---|---|---|---|---|
| | 740 | 760 | 780 | 800 | 820 |
| 0.0005 | 0.97618 | 0.99719 | 1.00238 | 1.02063 | 1.07884 |
| 0.001 | 1.00773 | 1.00260 | 0.97923 | 0.99909 | 1.23495 |
| 0.01 | 0.97764 | 0.94452 | 0.96002 | 0.91924 | 0.93990 |
| 0.1 | 0.93117 | 0.89593 | 1.03672 | 0.97166 | 0.90613 |
| 1 | 0.91977 | 0.93934 | 0.97871 | 0.91032 | 0.99239 |
| 10 | 1.13818 | 1.05228 | 1.03347 | 1.02899 | 1.02356 |

**Table 4.** The values of the material constant, $\zeta_2$, under all of the tested specified loading conditions.

| Strain Rate (s$^{-1}$) | Deformation Temperature (°C) | | | | |
|---|---|---|---|---|---|
| | 740 | 760 | 780 | 800 | 820 |
| 0.0005 | 0.99305 | 0.99204 | 0.99627 | 1.05767 | 0.98968 |
| 0.001 | 0.99713 | 0.98427 | 1.10590 | 1.01442 | 0.95979 |
| 0.01 | 0.97144 | 0.91887 | 0.95548 | 0.88622 | 0.91143 |
| 0.1 | 0.94142 | 0.98570 | 0.94737 | 0.90197 | 0.92483 |
| 1 | 0.86421 | 0.90325 | 0.82194 | 0.93762 | 0.85737 |
| 10 | 0.96838 | 0.96015 | 0.97821 | 0.98048 | 0.95020 |

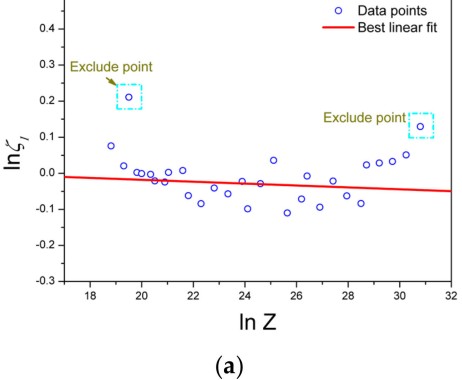
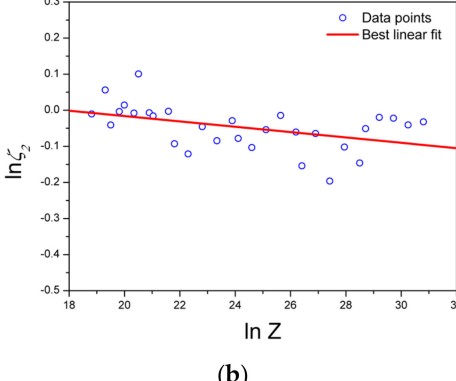

(a)  (b)

**Figure 9.** Relation between the material constant, $\zeta$, and the Zener-Hollomon parameter under (**a**) the prior-peak stage; and (**b**) the post-peak stage.

## 4.5. Verification of the Modified Constitutive Model

According to Section 4.4 and the already determined parameters of the OC model, the MC model, which is capable of predicting the flow behavior of Ti-10-2-3 alloy under all tested deformation stages, can be summarized as:

$$
\begin{cases}
\dfrac{\sigma}{\sigma_p} = \left[ \dfrac{\varepsilon}{\varepsilon_p} \exp\left(1 - \dfrac{\varepsilon}{\varepsilon_p}\right) \right]^c \\[2mm]
\sigma_p = \dfrac{1}{0.01044} \ln\left\{ \left(\dfrac{Z}{2.2035\times10^{10}}\right)^{\frac{1}{3.26}} + \left[ \left(\dfrac{Z}{2.2035\times10^{10}}\right)^{\frac{2}{3.26}} + 1 \right]^{\frac{1}{2}} \right\} \\[2mm]
\varepsilon_p = (5.818\times10^{-4})Z^{0.12} \\[2mm]
Z = \dot{\varepsilon}\exp\left(\dfrac{240005}{RT}\right) \\[2mm]
c = \begin{cases} 0.071Z^{0.0903}\,(\varepsilon \le \varepsilon_p) \\ (4.86\times10^{-5})Z^{0.1279}\,(\varepsilon > \varepsilon_p) \end{cases} \\[2mm]
\zeta = \begin{cases} 1.035Z^{-0.00263}\,(\varepsilon \le \varepsilon_p) \\ 1.141Z^{-0.00741}\,(\varepsilon > \varepsilon_p) \end{cases}
\end{cases}
\tag{21}
$$

With the modified constitutive model, the flow curve of the studied alloy during isothermal deformation can be predicted under a wide working window. The comparisons between the predicted flow stress from the MC model and the corresponding experimental ones are shown in Figure 10. The predicted data are in good agreement with the experimental results, which indicates that the employed parameter, $\zeta$, successfully compensates the error of the peak and plateau induced by the yield jump. Therefore, the MC model possesses great predictability that can describe the non-linear relation among flow behaviors, deformation temperature, strain rate, and strain of Ti-10-2-3 alloy.

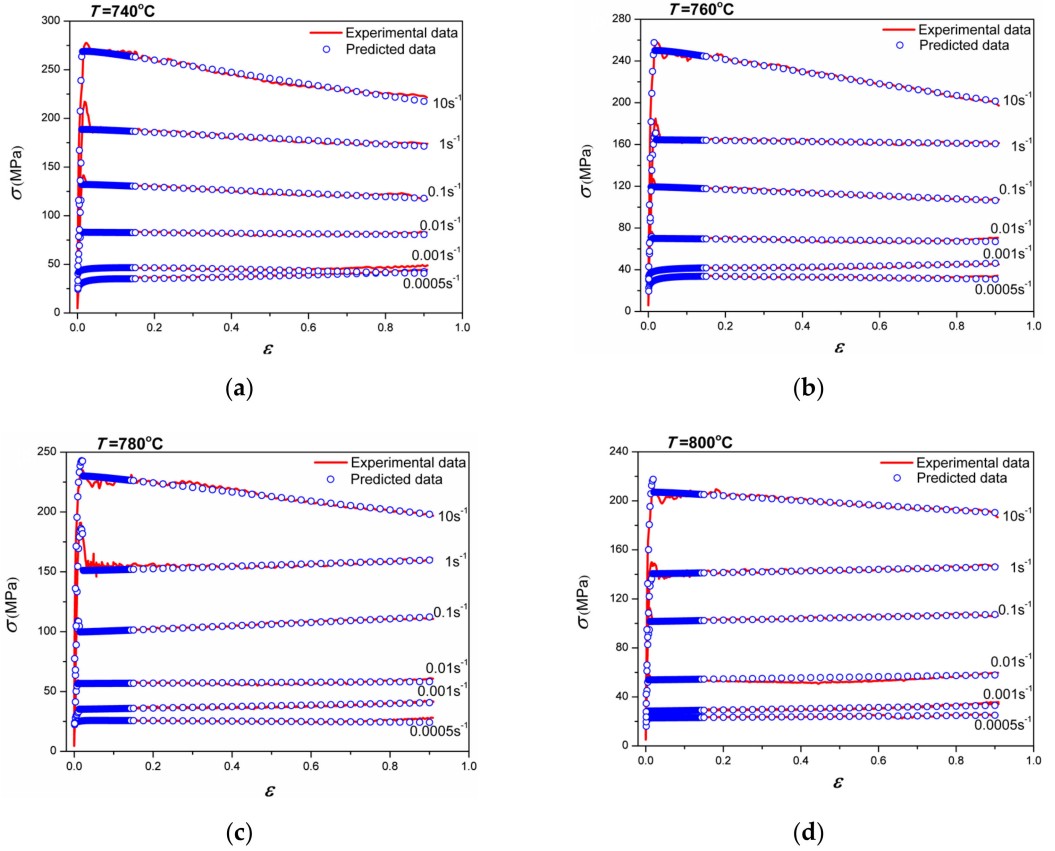

**Figure 10.** *Cont.*

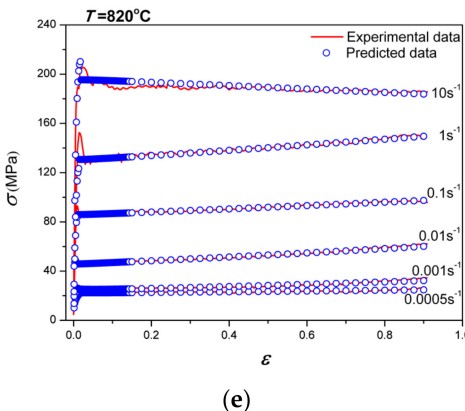

(**e**)

**Figure 10.** Comparisons between the experimental and the predicted flow stress by the MC model under the deformation temperatures of: (**a**) 740 °C; (**b**) 760 °C; (**c**) 780 °C; (**d**) 800 °C; and (**e**) 820 °C.

### 4.6. Comparison of the Original and the Modified Constitutive Models

In order to compare and analyze the predictability of the two models, the correlation coefficient, *R*, and the average absolute relative error, AARE, were used as the standard statistical parameters. *R* represents the strength of linear relation between the predicted and experimental values. However, in some cases, a higher value of *R* may not indicate a better model because the tendency of the scattered points may be biased towards a higher or lower value. Thus, it is necessary to calculate the AARE by comparing the relative error on a term-by-term method, which is an unbiased statistical parameter to evaluate the accuracy of the developed models [28]. The correlation coefficient and the average absolute error are defined as:

$$R = \frac{\sum_{i=1}^{N} (E_i - \overline{E})(P_i - \overline{P})}{\sqrt{\sum_{i=1}^{N} (E_i - \overline{E})^2 \sum_{i=1}^{N} (P_i - \overline{P})^2}}, \tag{22}$$

$$\mathrm{AARE}(\%) = \frac{1}{N} \sum_{i=1}^{N} \left| \frac{E_i - P_i}{E_i} \right| \times 100, \tag{23}$$

where $E_i$ denotes the experimental value, $P_i$ represents the predicted value obtained from the model, and $\overline{E}$ and $\overline{P}$ are the average values of $E$ and $P$, respectively. Further, $N$ is the total number of data sets employed in this investigation.

The correlation between the experimental flow stresses and the predicted values obtained from the OC and MC models in the prior- and post-peak stage are shown in Figure 11. For the OC model, the correlation coefficient, *R*, in the prior-peak stage is 0.99724, which is larger than that in the post-peak stage (0.97728). The value of the AARE is the opposite (Figure 11a,b), which indicates that the predictability of the OC model in stage I is better than in stage II. Because stage I, with a small strain, is a relative simple flow process, though it mainly includes work-hardening, dynamic recovery, and the initial dynamic recrystallization periods, it presents a nearly linear relation between stress and strain. It is easy for the OC model to describe the flow behavior under stage I. However, the flow behavior under stage II is difficult to describe because of a complex deformation mechanism, such as the discontinuous yield and the flow instability phenomena. Ji et al. [33] observed the deviation between experimental flow stresses and predicted ones from the constitutive equations in instability regimes. The values of *R* for the MC model under the prior-peak stage and the post-peak stage can be separately found as 0.99923 and 0.99958 (Figure 11c,d), respectively, which are larger than that of the OC model. Correspondingly, the values of the AARE for the MC model decreased significantly from 5.61% and 6.57% (OC model) to 2.86% and 1.28%.

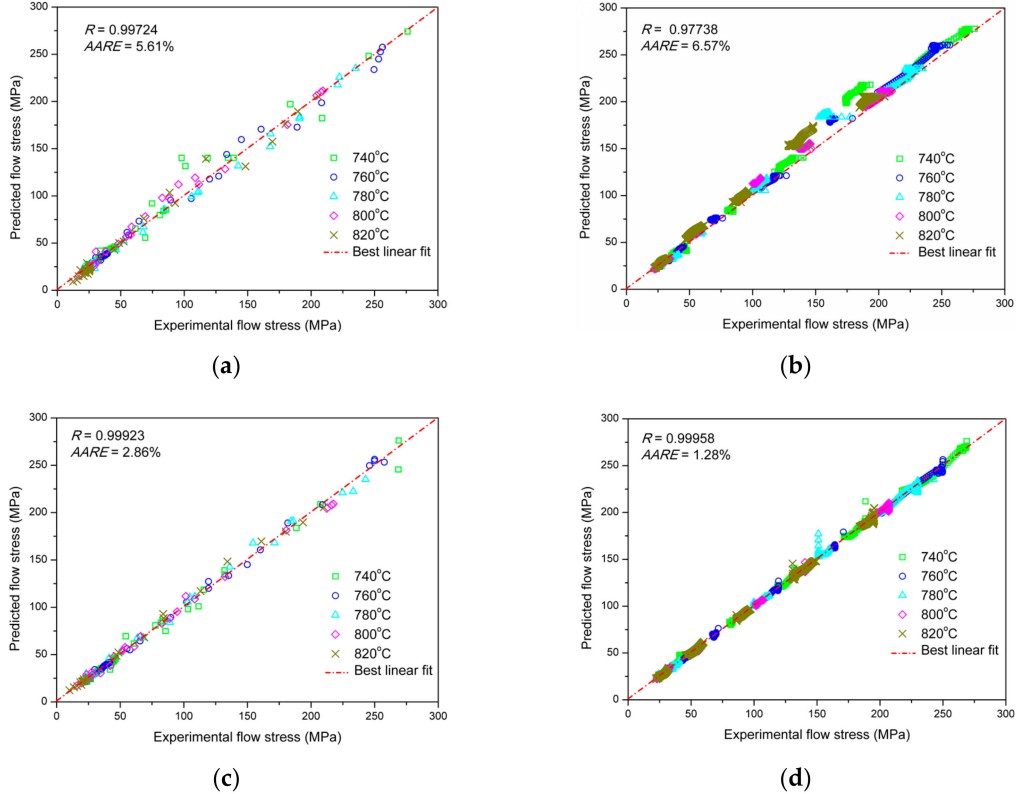

**Figure 11.** Correlations between the experimental flow stress and the predicted flow stress obtained from the OC model (**a**) and (**b**); MC model (**c**) and (**d**). Here, (a) and (c) represent the flow stress under the prior-peak stage; (b) and (d) denote that of the post-peak stage.

For comprehensive comparison, the values of *R* for the OC and MC models are 0.96735 and 0.99912, respectively (Figure 12). In addition, the AARE of OC model is 6.48%, much larger than the value of 1.44% for the MC model. Therefore, the flow behaviors established by the MC model have a higher accuracy and applicability for the Ti-10-2-3 alloy than the OC model.

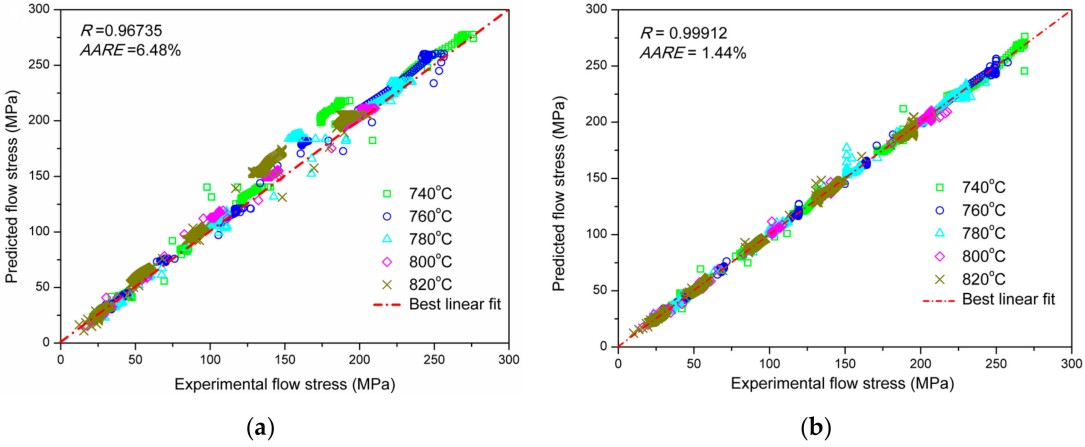

**Figure 12.** Correlations between the experimental flow stress and the predicted flow stress obtained from the (**a**) OC model and (**b**) MC model under all deformation conditions.

## 5. Discussion

Many investigations have used the OC model to predict the flow behavior of metallic materials with satisfactory results. However, most of these materials, like homogenized austenitic stainless

steels 301, 304, and 317 [24–26], are not that sensitive to the processing parameters. Hence, the OC model can predict the flow curve for a wide range of peak shapes. Further, the tested stress-strain curve is relatively smooth and does not show a severe yield jump. For materials that are sensitive to deformation parameters, the experimental flow curve rises to a sharp peak at small strains and then drops rapidly to a nearly steady value, presenting a discontinuous yield feature. Further investigation is necessary to explore the mechanism behind the discontinuous yield phenomenon. However, the developed MC model is able to predict the yield jump by an additionally introduced parameter. The additional parameter does not increase the calculation cost, since it can be simply determined from the *y*-intercept of the regression line in the plot of $\ln(\sigma/\sigma_p)$-$[\ln(\varepsilon/\varepsilon_p) + (1 - \varepsilon/\varepsilon_p)]$ when evaluating the original parameter, *c*, from the slope of that. Noticeably, compared with the OC model, the developed MC model can not only be applied in the shape of the flow curve with a broad peak but also significantly increases the prediction accuracy for flow curves with a sharp peak at small strain and followed by a rapid dropping to the plateau.

## 6. Conclusions

In this work, the hot flow behavior of Ti-10-2-3 alloy was studied by means of an isothermal compression test over the temperature range of 740 to 820 °C and the strain rate range of 0.0005 to 10 s$^{-1}$. The following conclusions can be drawn:

The studied alloy is extremely sensitive to deformation parameters, like the temperature and strain rate. The temperature mainly affects the magnitude of the flow stress at larger strains, while the strain rate not only affects the value of flow stress but also significantly determines the shape of flow curves. At low strain rates, the flow stress increases with the strain to a broad peak, followed by a nearly constant value of the flow stress. However, at high strain rates, the flow curves exhibit a hardening to a sharp peak at small strain, followed by a rapid dropping to a plateau.

Based on the experimental data, a phenomenological constitutive model for the description of the flow behavior was developed by introducing a new parameter into an existing model. The parameter was introduced in order to narrow the deviation of the predicted flow curves with the experimental ones. The introduced parameter, $\zeta$, did not add to the computational effort when accessing the developed constitutive equation.

Overall, comparisons between the OC model and the MC model indicated that the *R* of the OC model is 0.96735, much smaller than the value 0.99912 of the MC model, while the value of AARE was found to be the opposite of this. Specifically, the *R* of the OC and MC models under stage I were separately found to be 0.99724 and 0.99923. Correspondingly, the values of AARE were 5.61% and 2.86%, respectively. While the values of *R* under stage II transformed from 0.97738 (OC) to 0.99958 (MC), the values of AARE sharply decreased from 6.57% to 1.28%.

The description of the constitutive relation of the MC model has higher accuracy and applicability for the Ti-10-2-3 alloy than the OC model, which was confirmed by the comparison results between the predicted and experimental data. Additionally, the developed MC model can be used in the numerical simulation of hot deformation and design as well as for optimizing the processing parameters of the studied titanium alloy.

**Author Contributions:** Methodology, F.W.; experiment, F.W. and Y.Z.; data analysis, F.W. and J.S.; validation, Y.N.; writing—original draft preparation, F.W. and J.S.; reviewing and editing, Y.Z.; funding acquisition, Y.N.

**Funding:** The work was financially supported by the National Natural Science Foundation of China (Grant No. 51775440) and Fundamental Research Funds for the Central Universities (Grant No. 3102018ZY005).

**Conflicts of Interest:** The authors declare no conflict of interest.

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
