# Peer review of "A Modified Constitutive Model for the Description of the Flow Behavior of the Ti-10V-2Fe-3Al Alloy during Hot Plastic Deformation"

_metals, doi:10.3390/met9080844_

Round 1
Reviewer 1 Report
Dear Authors,
Thank you very much for your manuscript. The reviewer's comments can be found in the attached pdf-file.
Kind regards

Author Response
Dear reviewer
Thank you for giving us the opportunity to revise our paper entitled «A Modified Constitutive Model for the description of the flow behavior of the Ti-10V-2Fe-3Al Alloy during Hot Plastic Deformation» (metals-548520). In the revised version, the reviewer’s comments and suggestions have been taken into account (the modifications are highlighted in red in the paper) and the details are given in the attached document “A Modified Constitutive Model for the description of the flow behavior of the Ti-10V-2Fe-3Al Alloy during Hot Plastic Deformation”. Here are some answers to some of your suggestions.
1. Response for point 1 in broad comments:
Considering the structure of the paper, I have separated the Chapter2 “Experiments and Theories” into Chapter2 “Experiments” and Chapter3 “Theories”.
2. Response for point 2 in broad comments:
I have pointed out that the modelling approach presented in the chapter 4.1 and 4.2 is
based on the existing constitutive model of Lin et al. (Ref. [28]) and Hajari et al. (Ref.[29]).
3. Response for point 1 in specific comments:
I have changed the title of the article to “A Modified Constitutive Model for the description of the flow behavior of the Ti-10V-2Fe-3Al Alloy during Hot Plastic Deformation”.
4. Response for point 2 in specific comments:
For this purpose, I referred to literature [4] and [5].
Finite element analysis technique has been extensively used as an effective numerical tool in forming process optimization and structural design. And the effectiveness of simulation results is remarkely affected by the reliability and accuracy of predictability of inputted constitutive model [4,5].
5. Response for point 3 in specific comments:
I have changed the references to Lin and Chen (Ref. [8])
Lin, Y.C.; Chen, X.M. A critical review of experimental results and constitutive descriptions for metals and alloys in hot working, Mater. Des. 2011, 32, 1733-1759.
6. Response for point 5 in specific comments:
The cylindrical specimens with a diameter of 8 mm and a height of 12 mm were machined from the wrought billet.
7. Response for point 10 in specific comments:
The flow curve at 760°C and 0.0005 s−1 is nearly identical to that at 740°C
8. Response for point 16 in specific comments:
n and n' are two different material constants. We can refer to Hajari et al. [29]
Hajari, A.; Morakabati, M.; Abbasi, S.M.; Badri, H. Constitutive modeling for high-temperature flow behavior of Ti-6242S alloy, Mater. Sci. Eng. A. 2017, 681, 103-113.
9. Response for point 24 in specific comments:
the values of n' and β are averaged. I have corrected it in the article.
10. Response for point 28 in specific comments:
The symbol in Eq (9) represents the strain rate from 0.0005s-1 to 10s-1, and you can get the average value of Q for different strain rates.
(Ref. [29])
Hajari, A.; Morakabati, M.; Abbasi, S.M.; Badri, H. Constitutive modeling for high-temperature flow behavior of Ti-6242S alloy, Mater. Sci. Eng. A. 2017, 681, 103-113.
11. Response for point 31 in specific comments:
The difference in the value of n is due to the different calculation methods. The constant n obtained from figure 2c is the average of the inverse of the slope of several lines. The constant n obtained from figure 2e is the result of a data fitting. So it's normal to get a slightly different result.
12. Response for point 39 in specific comments:
An interesting phenomenon can be observed from Figure 8 that the peak's shape has little effect on the predictability of the constitutive model in the prior-peak stage (whether the shape of flow curve exhibits a sharp peak or a broad peak), while the predictability of the constitutive model in the post-peak stage is significantly rely on the peak’s shape, namely, the deviation increases as the fall from the peak to the steady state increases. In this case, the prediction error induced by the difference between peak and plateau is inevitable. Obviously, the value of the intercept in ln(σ/σp)-[ln(ε/εp) + (1 − ε/εp)] plot is smaller than zero, which doesn't consider the real value of intercept in Figure 6 (take the real intercept value as 0). This causes the value of C to be greater than the true value. So the predicted values are greater than the experimental values, as shown in Figure 8. Hence, the idea is to introducing a new parameter into Equation (2), which helps to minimize the error between the predicated and the experimental data.
Thank you very much and looking forward to hearing from you.
Best Regards,
Sincerely yours
Fukang Wang

Reviewer 2 Report
The manuscript entitled “A Modified Constitutive Model of the Ti-10V-2Fe-3Al Alloys during Hot Plastic Deformation Process” presents a modified constitutive model based on the original equation by introducing a new parameter to compensate the error between the experimental data and predicted ones. The manuscript can be considered for publication in Metals, but subject to a minor revision. By looking at the Fig. 10, it is clear that the modified models are not able to predict the ultimate strength at high strain rate, e.g., 1 s^-1. Can the authors explain why this occurs and if they have any suggestion to further improve the models?
Author Response
Dear reviewer
Thank you for giving us the opportunity to revise our paper entitled «A Modified Constitutive Model for the description of the flow behavior of the Ti-10V-2Fe-3Al Alloy during Hot Plastic Deformation» (metals-548520). In the revised version, the reviewer’s comments and suggestions have been taken into account (the modifications are highlighted in red in the paper) and the details are given in the attached document “A Modified Constitutive Model for the description of the flow behavior of the Ti-10V-2Fe-3Al Alloy during Hot Plastic Deformation”. Here are my answer for your suggestions.
For austenitic alloys, the flow curve transitions smoothly from initial loading to peak interval, the model will have a good prediction. However, it is very common for titanium alloys to appear the discontinuous yield after work hardening, the rheological curve usually produces the discontinuous region where the curve drops sharply from the peak to the lower yield point. Severe stress jitter occurs in a teeny strain interval. At this stage, it is difficult to obtain satisfactory prediction results for any model.
Thank you very much and looking forward to hearing from you.
Best Regards,
Sincerely yours
Fukang Wang

Round 2
Reviewer 1 Report
Dear Authors,
thank you very much for the correction of your manuscript and the detailed comments. From a technical point of view, the paper can be published in its present form. However, I strongly recommend a further revision of the English in order to improve the quality of your manuscript. You'll find some grammatical suggestions which will help to improve the English in the attached file.
Kind regards

Author Response
On behalf of my co-authors, we thank you very much for giving us an opportunity to revise our manuscript, we appreciate editor and reviewers very much for their positive and constructive comments and suggestions on our paper entitled «A Modified Constitutive Model for the description of the flow behavior of the Ti-10V-2Fe-3Al Alloy during Hot Plastic Deformation» (metals-548520). In the revised version, the reviewer’s comments and suggestions have been taken into account (the modifications are highlighted in red in the paper) and the details are given in the attached document “A Modified Constitutive Model for the description of the flow behavior of the Ti-10V-2Fe-3Al Alloy during Hot Plastic Deformation”. Here are some answers to some of your suggestions.
Response to Reviewer 1 Comments
1. Response for point 1 in broad comments:
We are very sorry for our negligence of the correlation coefficient (R-value) of Figure 2e, Figure 3, Figure 4, Figure 5, Figure 6 and Figure 7. All of those values have been added to the paper.
2. Response for point 6 in specific comments and grammatical suggestions
Considering the structure of the paper, a new section has been established in page 1.
3. Response for point 22 in specific comments and grammatical suggestions
To make it easier for readers to understand, I have shifted this two sentences to chapter3.
4. Response for point 26 in specific comments and grammatical suggestions
It is really true as reviewer suggested, and I have shifted this sentences to chapter 4.3.
5. Response for point 30 in specific comments and grammatical suggestions
After a few careful checks, we have made correction according to the reviewer’s comments. The unit of Q is J*mol-1.
6. Response for point 33 in specific comments and grammatical suggestions
We are very sorry for our incorrect writing of the material constants B and C.
We can refer to the chapter 3.2.3 Geng et al.
Geng, P.H.; Qin, G.L.; Zhou, J.; Zou, Z.D. Hot deformation behavior and constitutive model of GH4169 superalloy for linear friction welding process. J. Manuf. Process. 2018, 32, 469–481.
Thank you very much and looking forward to hearing from you.
